# Gut Bacteria Provide Genetic and Molecular Reporter Systems to Identify Specific Diseases

**DOI:** 10.3390/ijms25084431

**Published:** 2024-04-17

**Authors:** Leon M. T. Dicks

**Affiliations:** Department of Microbiology, Stellenbosch University, Stellenbosch 7600, South Africa; lmtd@sun.ac.za

**Keywords:** gut microbiota, reporter systems, diseases

## Abstract

With genetic information gained from next-generation sequencing (NGS) and genome-wide association studies (GWAS), it is now possible to select for genes that encode reporter molecules that may be used to detect abnormalities such as alcohol-related liver disease (ARLD), cancer, cognitive impairment, multiple sclerosis (MS), diabesity, and ischemic stroke (IS). This, however, requires a thorough understanding of the gut–brain axis (GBA), the effect diets have on the selection of gut microbiota, conditions that influence the expression of microbial genes, and human physiology. Bacterial metabolites such as short-chain fatty acids (SCFAs) play a major role in gut homeostasis, maintain intestinal epithelial cells (IECs), and regulate the immune system, neurological, and endocrine functions. Changes in butyrate levels may serve as an early warning of colon cancer. Other cancer-reporting molecules are colibactin, a genotoxin produced by polyketide synthetase-positive *Escherichia coli* strains, and spermine oxidase (SMO). Increased butyrate levels are also associated with inflammation and impaired cognition. Dysbiosis may lead to increased production of oxidized low-density lipoproteins (OX-LDLs), known to restrict blood vessels and cause hypertension. Sudden changes in SCFA levels may also serve as a warning of IS. Early signs of ARLD may be detected by an increase in regenerating islet-derived 3 gamma (REG3G), which is associated with changes in the secretion of mucin-2 (Muc2). Pro-inflammatory molecules such as cytokines, interferons, and TNF may serve as early reporters of MS. Other examples of microbial enzymes and metabolites that may be used as reporters in the early detection of life-threatening diseases are reviewed.

## 1. Introduction

The vast majority of human microbiota (97%) is found in the gastrointestinal tract (GIT), largely in the colon [1,2], and includes bacteria, archaea, fungi, and viruses [3]. The more than three million genes of gut microbiota (all phyla included) supersede the estimated 23,000 genes in the human genome, which emphasizes the role the gut microbiome plays in human health [4]. Bacterial numbers, more or less equal to the number of human cells, are the best studied [2,4], and approximately 90% are grouped into the phyla Firmicutes (Gram-positive) and Bacteroidetes (Gram-negative), with the rest belonging to the phyla Actinobacteria, Fusobacteria, Proteobacteria, and Verrucomicrobia [4,5,6]. An estimated 60–80% of gut bacteria have not been isolated and their phenotypic properties have not been studied, mainly because they are unculturable [7,8]. This implies the role many species play in the gastrointestinal tract (GIT), specifically regarding gut homeostasis, regulation of the immune system, maintenance of intestinal epithelial cells (IECs), and the regulation of neurological and endocrine functions are unknown [3].

With techniques such as 16SrRNA sequencing, next-generation sequencing (NGS), and genome-wide association studies (GWAS), many of the unculturable microorganisms have been identified and their function in the gastrointestinal tract (GIT) is deduced from gene expressions. The genetic information allows for the design of genetic probes to determine the presence of species in the GIT and select or design target molecules to detect the early start of diseases such as alcohol-related liver disease (ARLD), cancer, cognitive impairment, multiple sclerosis (MS), diabesity, and ischemic stroke (IS). Common gastrointestinal diseases directly linked to dysbiosis, e.g., irritable bowel syndrome (IBS) [9,10], delayed gastric emptying in the absence of mechanical obstruction, referred to as gastroparesis (GP) [11], and endometriosis (EM) [12] may also benefit from the development of reporter molecules.

Individuals suffering from IBS are often diagnosed with depression and anxiety [9]. A meta-analysis of 777 patients diagnosed with IBS has shown that these individuals have higher levels of Firmicutes and lower levels of Bacteroidetes in their fecal microbiota compared to non-IBS patients [13]. The authors also reported an increase in clostridia and Clostridiales, but a decrease in Bacteroidia and Bacteroidales. A separate meta-analysis performed on 1340 IBS patients revealed lower levels of *Lactobacillus* and *Bifidobacterium*, and higher levels of *Escherichia coli* and *Enterobacter* in fecal samples compared to healthy subjects [14]. The authors found no significant differences in the levels of fecal *Bacteroides* and Enterococcus. Attempts to alleviate IBS with probiotic supplements and changes in diet have not always been effective. Treatment with antibiotics such as rifaximin has alleviated symptoms of IBS [15,16], but this was accompanied by a modest, largely transient effect across a broad range of gut microbiota [17]. Fecal microbiota transplants (FMTs) also have not provided prolonged relief from IBS [9].

Symptoms of GP are similar to those of other functional diseases, especially functional dyspepsia (FD) [11]. The epidemiology of GP is thus difficult to ascertain. This is further complicated by relatively few cases reported, e.g., approximately 24 per 100,000 individuals in the USA and 14 per 100,000 in the UK [18,19]. GP and FD are associated not only with an increase in bacterial overgrowth in the duodenum but also with alterations in microbial composition. Analyses of the composition of duodenal mucosal-associated microbiota (MAM) in FD patients have revealed a dominance of *Streptococcus* spp. but lower cell numbers of anaerobic *Prevotella*, *Veillonella*, and *Actinomyces* [20].

Endometriosis is a chronic, estrogen-dependent inflammatory condition characterized by the presence of endometrium-like tissue outside the uterus [21]. An estimated 10% of women in their reproductive years are affected by EM, corresponding to approximately 190 million women worldwide [21]. As one study shows, individuals diagnosed with EM had increased levels of *Proteobacteria*, *Enterobacteriaceae*, *Streptococcus* spp., and *Escherichia coli* [22]. Conflicting results were reported by the authors for Firmicutes and *Gardnerella* spp. Independent studies have found significant increases in *Actinobacteria*, *Cyanobacteria*, *Saccharibacteria*, *Fusobacteria*, *Acidobacteria*, and *Patescibacteria* in patients suffering from EM [23,24,25]. Notable increases have been reported in cell numbers of *Proteobacteria*, *Bacteroidetes*, and *Negativicutes*, particularly *Shigella*, *E. coli*, and *Prevotella* [26,27].

The development of reporter systems to diagnose early signs of bacterial-inflicted diseases requires a thorough understanding of conditions that influence the expression of genes and the production of bacterial metabolites (defined as small molecules released by bacteria), short-chain fatty acids (SCFAs), tryptophan, hormones such as serotonin (5-HT), cholecystokinin (CCK), and peptide tyrosine-tyrosine (PYY), as well as neurotransmitters such as glutamate (Glu), γ-aminobutyric acid (GABA), dopamine (DA), norepinephrine (NE), and histamine (His), in addition to the role all these compounds play in communication with the central nervous system (CNS) via the gut–brain axis (GBA), hypothalamic pituitary adrenal axis (HPA), and autonomic sympathetic and parasympathetic nervous systems [28,29,30]. Afferent signals generated by gut microbiota reach the CNS via the enteric nervous system (ENS) which is in close contact with the gut wall of the circulatory system. Efferent signals from the brain are communicated to gut microbiota via the ENS. The ENS is often referred to as the “brain within the gut” or “second brain” [31] and is a dynamic and ever-changing network of neurons, maintained by multiple apoptotic and neurogenetic processes that are regulated by gut microbiota, especially those with the ability to produce and metabolize SCFAs, long-chain fatty acids (LCFAs), and hormones [32]. Minor changes in the gut microbiome alter the type and level of neurotransmitters produced, which in turn affects digestion, intestinal permeability, gastric motility, and immune regulation [4]. SCFAs play a major role in gut homeostasis but are equally important in the maintenance of intestinal epithelial cells (IECs), regulation of the immune system, and neurological and endocrine functions [30]. Changes in SCFA levels may serve as an early warning of colon cancer, inflammation, cognitive abnormalities, and IS. Changes in the secretion of mucin-2, brought about by changes in the population of gut microbiota, lead to an increase in regenerating islet-derived 3 gamma (REG3G), which serves as an early reporter of ARLD [7].

This review looks at the possibility of developing disease-specific genetic- or molecular-based reporter systems produced by gut microbiota that may be used to identify abnormalities at an early stage. Only a select few diseases with the most advanced research on therapeutic targets are discussed. The purpose of the review is not to discuss the diseases in detail but rather to identify possible reporter molecules that may be used in early detection.

## 2. Alcohol-Related Liver Disease (ARLD)

Moderate consumption of alcohol raises levels of high-density lipoprotein (HDL), referred to as “good cholesterol” and is associated with protection against heart disease [33]. However, excess alcohol intake may lead to the development of alcohol-related liver disease (ARLD), which contributes to 47.9% of all liver cirrhosis deaths [33]. Severe alcoholic hepatitis, the most aggressive form of ARLD, has a 30-day mortality rate ranging from 20 to 50% [34]. Symptoms vary from feeling sick to weight loss, loss of appetite, yellowing of the whites of the eyes or skin (jaundice), swelling of ankles and midsection of the body, drowsiness, vomiting of blood, and blood in the stool. The diagnosis of ARLD is challenging and in most cases relies on the history of the patient and clinical and laboratory findings [35]. Blood analyses are usually performed (summarized in Figure 1) and, amongst other tests, include testing for the levels of transaminases, bilirubin, alkaline phosphatase (ALP), albumin, gamma-glutamyl transferase, and prothrombin. The latter test (prothrombin time or PT) is to determine the blood clotting rate (referred to as the international normalized ratio, INR, blood test) [35]. Biochemical markers used to detect chronic ARLD are gamma-glutamyl transferase (GGT), aspartate aminotransferase (AST), alanine aminotransferase (ALT), mean corpuscular volume (MCV), and carbohydrate-deficient transferrin (CDT) [36,37]. An AST:ALT ratio above two is normally considered a reliable indication of ARLD [38]. However, patients with ARLD may not always have elevated serum aminotransferase levels [39]. Further confirmation of ARLD is an increase in the ratio of immunoglobulin A (IgA) versus IgG [40]. No single biomarker is sensitive and specific enough to confirm ARLD. However, by using a combination of biomarkers such as CDT, GGT, and MCV, the sensitivity of the test increases significantly. In many cases, CDT testing is combined with screening questionnaires. In severe cases, liver biopsy is performed to detect hepatic steatosis, inflammation, and Mallory–Denk bodies [35].

Since alcohol dependency is not only regulated by the brain but also by gut microbiota [41], it is safe to conclude that the GBA plays a decisive role in ARLD. Individuals diagnosed with ARLD show clear changes in populations of *Lactococcus*, *Streptococcus*, *Psychrobacter*, *Helicobacter*, *Alloprevotella*, *Paenalcaligenes*, and *Janthinobacterium* compared to non-alcoholics (Figure 2) [42]. In patients diagnosed with Crohn’s disease, inflammation is associated with a drastic decrease in *Faecalibacterium prausnitzii* [43]. Similar results have been reported when mice were exposed to a high-alcohol diet [44]. The high levels of butyrate produced by *F. prausnitzii* stimulate the expression of tight junction proteins and the production of mucin, all of which are deemed important in maintaining the integrity of the gut wall. With the depletion of *F. prausnitzii* the individual with ARLD is at risk of developing a leaky gut.

Diet plays a major role in selecting gut microbiota and the protection of intestinal epithelial cells (IEC), as shown by the increase in bifidobacteria with a diet rich in inulin-type fructans [45]. Bifidobacteria are proliferating butyrate producers but are sensitive to alcohol. It is thus no surprise that metagenomic studies conducted on mice exposed to excessive alcohol intake have shown a decrease in the production of butyrate and saturated long-chain fatty acids (LCFAs) [46]. Since *Lactobacillus* spp. use saturated LCFAs as an energy source, their cell numbers are also drastically reduced in individuals with ARLD. This leads to further disruption of tight junctions. Feeding mice with impaired gut walls saturated LCFAs has led to the recovery of tight junctions, an increase in *Lactobacillus* cell numbers, and an alleviation of symptoms associated with ARLD [46]. Supplementation of diets with LCFAs must, however, be carefully controlled, as it may lead to the overgrowth of *Lactobacillus* spp., dysbiosis, and hepatic steatosis (fatty liver disease) [47,48]. Screening for overall changes in the microbial population, SCFAs (e.g., butyrate), and LCFAs is not the answer to detecting early signs of ARLD as cell numbers, species, and nutrient compositions vary with changes in diet, stress, and medication [3,4,49]. However, since *F. prausnitzii* and bifidobacteria are strongly associated with butyrate production, genes encoding enzymes in butyrate pathways may hold the answer to the development of DNA probes that can be used as reporters of early ARLD.

Dysregulation of bile acid (BA) metabolism is typical of individuals diagnosed with ARLD. This leads to the malfunctioning of tight junction proteins, followed by intestinal inflammation [46,50] and the formation of reactive oxygen species that disrupt cell membranes and mitochondria [51]. Lithocholic acid (LCA), a derivative of cholic acid, induces stress on the endoplasmic reticulum (ER) and enhances cell death (Figure 3), which is described as “unfolded protein response (UPR)-activating cell death” [52]. LCA transforms growth factor-β in HepG2 liver cancer cells, induces oxidative phosphorylation (Figure 3), stimulates antitumor immunity, and inhibits epithelial–mesenchymal transition and the expression of vascular endothelial growth factor A [53]. Changes in BA may thus serve as an early indication of ARLD and the health status of IECs.

A diet rich in proteins and low in carbohydrates may lead to an increase in bile-producing *Clostridium scindens*, *Clostridium hiranonis*, *Clostridium hylemonae*, and *Clostridium sordellii* [54]. Since these species are closely associated with colorectal cancer (CRC) [55], a species-specific diagnostic test is invaluable, provided that the tests are performed on a routine basis and diets remain unchanged. Bile acids serve as ligands for farnesoid X receptor (FXR) and induce the FXR target gene fibroblast growth factor (FGF)-19 in humans. The release of FGF-19 into the portal vein inhibits the synthesis of hepatic BA in the liver [56]. FGF-19 may thus serve as a reporter molecule for alcoholic hepatitis and early ARLD.

Another possible reporter molecule for ARLD is the c-type lectin called regenerating islet-derived 3 gamma (REG3G). Studies conducted on mice with ARLD have shown that the increase in mucosal bacteria, especially in the jejunum, is associated with a decline in REG3G production [57]. Mice deficient in the *Reg3g* gene developed alcoholic steatohepatitis [58], confirming the protective role of REG3G in ARLD. An increase in REG3G prevented ARLD in mice [58]. The authors have also shown an increase in inflammation and steatosis in REG3G-deficient mice, associated with the translocation of bacteria to mesenteric lymph nodes and the liver. Bacteria-derived products such as lipopolysaccharides (LPS) and fungal cell wall components are also translocated across impaired gut walls (Figure 3) [59]. The overexpression of *Reg3g* in mice led to a decrease in mucosa-associated microbiota and prevented the translocation of bacteria, thus protecting the animals from developing alcoholic steatohepatitis [58]. Studies on humans have yielded similar results. Furthermore, a deficiency of mucin-2 (Muc2), the most abundantly secreted mucin in the small and large intestine, resulted in an increase in antimicrobial lectins, including REG3B and REG3G [60,61]. REG3B and REG3G may serve as reporter molecules to detect early changes in Muc2, which is strongly associated with ARLD.

Most studies have focused on ARLD. Less information is available on non-alcoholic fatty liver disease (NAFLD) and non-alcoholic steatohepatitis (NASH). High levels of fructose increase intestinal permeability due to the loss of tight junction proteins. This, in turn, leads to an increase in the translocation of endotoxins, damage to hepatocytes, the activation of Kupffer cells, and the release of inflammatory cytokines and oxygen radicals [62]. These changes stimulate Toll-like receptors (TLRs) in the liver [63] and lead to increased levels of LPS. Individuals with NAFLD have lower cell numbers of *Bacteroides* and *Bifidobacterium*, but increased numbers of *Clostridium* and *Ruminococcaceae* [64,65]. Similar results were reported by Xue et al. [66] when they studied rats with NAFLD, except that the authors reported an increase in *E. coli* and *Enterococcus* spp., and a decrease in *Lactobacillus*, *Bifidobacterium*, and *Bacteroides*.

Studies with mice have shown that alcohol abuse may lead to an increase in the expression of the *Per2* gene that encodes the period circadian regulator Per2. This led to the dysfunction of occludin in IECs of the duodenum and colon, followed by the translocation of bacteria across the gut wall (Figure 3) [67]. Whether this applies to humans is uncertain. According to Leclercq et al. [41] only 43% of individuals diagnosed with alcohol use disorder and mild liver disease showed evidence of gut barrier dysfunction when tested using the 51Cr-EDTA method. Further research is required to determine if Per2 or its gene *Per2* could be used as reporters of bacteremia due to alcohol misuse.

Individuals with liver cirrhosis have high levels of LPS, tumor necrosis factor-α (TNFα), endotoxin, and pathogen-associated molecular patterns (PAMPs), usually associated with an increase in bacteroidetes, proteobacteria, *Enterobacteriaceae*, *Alcaligenaceae*, *Fusobacteriaceae*, *Prevotellaceae*, bifidobacteria, and streptococci, and a decrease in *Lachnospiraceae*, *Ruminococcaceae*, *Coprococcus*, and *F. prausnitzii* [50]. Gut microbiota most inhibited by alcohol are *Clostridium*, *Akkermansia*, and *Bacteroides*, and those stimulated are *Enterobacteriaceae*, *Actinibactaeria, Proteobacteria, Firmicutes, Prevotellaceae*, *Veillonellaceae*, and *Streptococcoceae* [68]. Alterations of these microbiota are associated with an increase in endotoxins and β-glucan, and a decrease in amino acids, steroids, SCFAs, LCFAs, indole-3 acetic acid, vitamin B, and bacteriocins [68].

More research is required to determine the effect of bacterial diversity on ARLD, and to what extent treatment with beneficial microorganisms could alleviate symptoms. Repair of the gut barrier is possible with microbial fecal transplants (MFTs), supplementation with SCFAs, and synthetic biology interventions [69]. Despite all the knowledge gathered on ARLD, we still lack a detailed understanding of which species are more prone to cause a leaky gut, the species or strains best suited to rectify damage to the gut wall, and the most harmful circulating PAMPs. Although changes in microbial populations have been identified, we have not yet developed a set of microbial markers that may be used as an early reporting system for liver disease. This is important, as currently available biomarkers for ARLD are not always sensitive enough, and not all patients display elevated serum aminotransferase levels.

## 3. Cancer

Biomarkers currently used to detect the most prevalent cancers (Figure 4) are alpha-fetoprotein (αFP) (testicular and hepatocellular cancer), human chorionic gonadotrophin (β-HCG) (testicular cancer), calciton and thyroglobulin (thyroid cancer), cancer antigen (CA) 125 (ovarian cancer), CA 19.9 (pancreatic cancer), CA 15.3 (breast cancer), carcinoembryonic antigen (CEA) (colorectal cancer), paraproteins (myeloma), and prostate-specific antigen (PSA) (prostate cancer) [70]. A better understanding of microorganisms associated with tumors and the carcinogenic metabolites they produce may, however, lead to the development of early cancer reporters. Some examples are discussed.

*Fusobacterium nucleatum* and *Helicobacter pylori* promote cancer through a complex set of mechanisms, including chronic inflammation, DNA damage, and the activation of oncogenic pathways [71,72]. Proteases and phospholipases produced by *H. pylori* degrade the mucus layer on the surface of gut epithelial cells. This not only facilitates the adherence of *H. pylori* to mucus layers [73] but also upregulates the cytotoxin-associated gene A (*CagA*), leading to an increase in spermine oxidase (SMO). During the conversion of polyamine spermine to spermidine, H_2_O_2_ levels increase, causing damage to the DNA of epithelial cells and apoptosis. A subpopulation of the affected epithelial cells gradually becomes resistant to apoptosis and transforms into malignant cells [74,75]. Infection with *H. pylori* is, however, not an indication of cancer. Only 1–4% of individuals infected with *H. pylori* develop gastric cancer. Colibactin, a genotoxin produced by polyketide synthetase-positive strains of *Escherichia coli* (pks^+^ cells), alkylates DNA and causes colorectal cancer [76,77]. Screening for colibactin-producing *E. coli* may be a strategy to restrain the production of pro-tumorigenic factors from the tumor microenvironment. Our understanding of the cancer-promoting potential of pks^+^ *E. coli* is limited and the expression of SMO in colorectal tumors needs to be confirmed with more clinical studies. In the future, colibactin and SMO may serve as reporters of colon cancer.

Microbial cells have also been also detected in various other tumors, including breast and pancreatic cancers [76,77]. Although it seems feasible to use bacteria as biomarkers to predict different types of cancer, differentiation between cancer-causing microbes within cells and those surrounding cancer cells or immune cells remains difficult [78,79,80]. This is further complicated by low cell numbers and the infrequent distribution of bacterial cells in tumors [81,82]. The average distribution of microorganisms in malignant tumors is one in every 10^4^ cells [83]. It is thus clear that microorganisms associated with cancer cells can only be detected with susceptible molecular methods that are specific enough to differentiate between bacterial species and detect the progression of bacterial cells in different tissues. Data generated by research groups that screen for microorganisms associated with cancer are inconsistent and unreliable. This may change with the vast strides made in the identification of invading species with next- and third-generation sequencing [84,85].

Some cancers are not caused by direct contact with microbial cells or TME but by microbial metabolites. Growing evidence indicates that commensal bacteria are involved in the pathogenesis and progression but also the suppression of various human cancers. Bacterial communities that populate solid tumors have been described. Some microbial molecules can be therapeutically exploited to detect cancer at an early stage. Examples are lipoteichoic acid (LTA) and deoxycholic acid (DCA) translocated from the GIT to the liver via enterohepatic circulation and the release of outer membrane vesicles (OMVs) from microbial cells [86,87]. OMVs released by *Fusobacterium nucleatum* subsp. *polymorphum* enter the bloodstream through capillaries and spread to distantly located cells. Activation of TLR4 and NF-κB in colonic epithelial cells stimulates the production of downstream pro-inflammatory factors that cause intestinal inflammation [88]. It is, however, difficult to detect low quantities of TLRs in the intestinal mucosa. Quantitative PCR has been used to detect the expression of specific genes but does not always provide consistent results due to variations of TLRs within the mucosa. Although most TLRs are expressed in IECs in the large intestinal tract, in vitro tests have shown that the expression of TLR and signaling in IECs is down-regulated [89]. On the other hand, in patients with inflammatory bowel disease (IBD), an increase in the expression of mucosa-located TLR4 has been reported [89]. Ungaro et al. [90]. used immunohistochemistry (IHC) instead and claimed that they could detect different expression levels of TLR4 in several cell types. By using immunofluorescence (IF) and IHC methods, the authors discovered that a subset of human colorectal cancer cells overexpressed TLR4. OMVs secreted from *F. nucleatum* alter epithelial homeostasis by targeting the receptor-interacting protein kinase 1 (RIPK1)-mediated cell death pathway. This weakens the intestinal mucosal barrier and leads to ulcerative colitis [91,92].

Melanoma patients treated with a combination of T-lymphocyte-associated protein 4 (CTLA-4) and programmed cell death protein 1 (PD-1) have developed immune-related adverse events (irAEs) of grade three or higher [93]. The binding of programmed cell death ligand-1 (PDL-1) to PD-1 blocks treatment with immunotherapy drugs. Treatment that targets the PD-1/PD-L1 axis has proved highly successful but is not without challenges. In many cases, blocking the PD-1/PD-L1 axis is not sufficient to stimulate an effective antitumor immune response, as seen in HCC patients. This suggests that the PD-1/PD-L1 pathway is not the only rate-limiting factor [94]. According to the authors of one study, the PD-1/PD-L1 antibody may be used in combination with antivascular drugs, dual immunotherapy, and combined with radiation and chemotherapy. *Coprobacillus cateniformis* has been shown to down-regulate PD-L2 expression on dendritic cells (DCs) and increase the efficacy of PD-1 inhibitors [95]. Research on the efficacy of drug therapy, drug reactions, and the identification of tumor biomarkers is a challenging field and requires in-depth fundamental research. The answer to the treatment of gastrointestinal cancer may very well be the selection of gut bacteria with anti-PD-1 and PD-L1 properties. Melanoma patients who received MFTs have defeated resistance to anti-PD-1 therapy [96,97].

*B. intestinalis* triggers the occurrence of irAEs by inducing the expression of IL-β1 in the ileum [93]. Studies conducted on mice have shown that SCFAs limit the activity of anti-CTLA-4 by restricting the up-regulation of CD80/CD86 on dendritic cells (DCs) and inducible costimulatory (ICOS) on T cells and cause the accumulation of tumor-specific and memory T cells [98]. Butyrate induces the differentiation of colonic regulatory T cells (Tregs) [99] and may thus suppress antitumor immunity. In another study conducted on patients diagnosed with HCC and treated with immune checkpoint inhibitors (ICIs) [100], cell numbers of *Lachnoclostridium* increased, and thus also the levels of ursodeoxycholic acid (UDCA) and ursocholic acid (UCA). Although several immunotherapy biomarkers, such as PD-L1, tumor mutational burden (TMB), and tumor-infiltrating T cells have been identified in different types of cancer [101,102,103], none have been validated clinically. These biomarkers are often difficult to manipulate, which limits their practical applications. Furthermore, innate and adaptive immunity could be regulated by gut microbiota and their metabolites [104].

In several studies, patients diagnosed with liver cancer who responded to immunotherapy had elevated numbers of *Lachnospiraceae*, *Alistipes*, *Marseille*, and *Ruminococcaceae* that could be associated with longer progression-free survival (PFS) and overall survival (OS). Patients who did not respond to immunotherapy had high numbers of *Veillonellaceae* and worse PFS and OS outcomes [105,106]. Lung cancer patients who reacted positively to immunotherapy had high levels of *Alistipes putredinis*, *Bifidobacterium longum*, *Bacteroides vulgatus*, *Prevotella copri*, and *Parabacteroides distasonis*, while patients who did not respond to immunotherapy had low numbers of *Ruminococcus* [107,108]. *Phascolarctobacterium* and *Ruminococcus* spp. are associated with improved prognosis in lung cancers, while an increase in *Dialister* spp. is linked to a shorter PFS [109,110]. *Bifidobacterium longum*, *Bifidobacterium adolescentis*, *Collinsella aerofaciens*, and *Enterococcus faecium* have been shown to be more abundant in patients with metastatic melanoma who responded to treatment compared to non-responding patients [111].

In general, Firmicutes are associated with a positive immunotherapy response in HCC patients, while Bacteroidetes are dominant in biliary tract cancer (BTC) patients who respond favorably to immunotherapy [105]. High cell numbers of *Bacteroides caccae* have been recorded irrespective of the type of ICI therapy. High cell numbers of *Faecalibacterium prausnitzii*, *Bacteroides thetaiotamicron*, and *Holdemania filiformis* have been observed in patients who responded to Ipilimumab and Nivolumab, while an increase in *Dorea formicogenerans* has been noted in patients treated with pembrolizumab [112]. *Bacteroides zoogleoformans* has been associated with improved responses to immunotherapy, while *Bacteroides ovatus*, *Bacteroides dorei*, and *Bacteroides massiliensis* have been associated with worse PFS [105,113]. Based on these results, variations in bacterial species, even within the same genus, can lead to opposite conclusions. Furthermore, results reported for the same type of cancer are not always consistent. It is clear that the selection of immunotherapy biomarkers is not that simple, and reports need to be critically evaluated. The inconsistency in results may be attributed to the variation of cancer types, analysis methods, sample size, immunotherapy drugs used, pretreatments, and clinical history of patients. A good approach to achieving consistent data would require the standardizing of trials, the enrollment of more participants, and the use of updated interdisciplinary methods.

HCC is the third most common cause of cancer-related fatalities globally and is influenced by the gut microbiome [114]. This means biomarkers based on gut microbiota may be used in the early detection of HCC [115,116]. In patients with NAFLD-HCC, gut microbiota stimulates IL-10^+^ Tregs, causes a decrease in pro-inflammatory cytokines such as IL-2 and IL-12, and leads to the attenuation of cytotoxic CD8^+^ T cells. According to Behary et al. [117] such changes in immunosuppression may contribute to the progression and development of NAFLD-HCC. In mice dysbiosis has been shown to increase the infiltration of myeloid-derived suppressor cells (MDSCs) to the liver, leading to liver carcinogenesis and a decrease in *Akkermansia muciniphila* in the GIT [118]. This leads to an increase in SCFAs [117]. Butyrate and propionate regulate CD4^+^ and CD8^+^ T cells and reduce inflammation [99,119]. Patients diagnosed with HCC have shown a decrease in butyrate-producing *Ruminococcus*, *Oscillibacter*, *Faecalibacterium*, *Clostridium* IV, and *Coprococcus* and an increase in LPS-producing bacteria such as *Klebsiella* and *Haemophilus* [120]. Primary bile acids that are converted to secondary bile acids by gut microbiota are reabsorbed in the intestine and promote hepatic inflammation and hepatocarcinogenesis [121,122,123]. In individuals with NAFLD-HCC and NAFLD-cirrhosis, *Proteobacteria*, *Enterobacteriaceae*, *Bacteroides caecimuris*, and *Veillonella parvula* are dominant. *Oscillospiraceae* and *Erysipelotrichaceae* are less prominent in NAFLD-HCC patients [117]. Less *Faecalibacterium*, *Ruminococcus*, and *Ruminoclostridium*, normally associated with SCFA production, have been isolated from the feces of patients who tested negative for hepatitis B virus and hepatitis C virus [non-B, non-C (NBNC) hepatitis] and (NBNC)-HCC. However, higher levels of pro-inflammatory bacteria such as *Escherichia-Shigella* and *Enterococcus* were identified in these patients [123].

Several studies have shown that gut dysbiosis may lead to the development of colorectal cancer (CRC) [124,125]. Accurate and noninvasive biomarkers for early CRC screening are needed. A starting point would be further research on oncogenic strains of *Bacteroides fragilis*, *Escherichia coli*, *Enterococcus faecalis*, *Streptococcus gallolyticus*, and *Fusobacterium nucleatum.* Although different datasets show a drastic change, mostly a reduction in species, in the microbiome of individuals with CRC [126,127], no single strain has been identified as a universal biomarker. *Fusobacterium nucleatum* may be used as a biomarker for CRC in feces and tumors [128,129,130], but further research is needed to link cell numbers to readings recorded with fecal immunochemical tests (FITs) and exposure to the anti-cancer drug. Another promising non-invasive approach to detect early CRC is the detection of microbial-derived metabolites in blood, urine, saliva, and fecal samples [131], possibly by using nuclear magnetic resonance (NMR) [132]. Target molecules would be lactate, glucose, and specific amino acids, as these were present at higher levels in CRC patients compared to healthy controls [133]. Changes in the levels of SCFAs, glutamate, and succinate may be used to follow the development of CRC, as these vary with the progression of tumors [132]. According to Lin et al. [132], acetate levels varied the most in the feces of CRC patients and are thus a good biomarker. Chen et al. [134], on the other hand, have noted that the levels of butyrate decreased in the stools of CRC patients as the numbers of butyrate-producing bacteria declined. Liu et al. [135] have shown that *Desulfovibrio*, *Escherichia*, *Faecalibacterium*, and *Oscillospira* may be used as fecal biomarkers, with a GC prediction of 90% and above.

## 4. Cognitive Impairment

The gut microbiome regulates the metabolism of D-amino acids in the brain and plays a major role in cognitive impairment [136,137,138], as depicted in Figure 5. *Corynebacterium glutamicum*, *Lactobacillus plantarum*, *Lactococcus lactis*, *Lactobacillus paracasei*, *Brevibacterium avium*, *Mycobacterium smegmatis*, *Bacillus subtilis*, and *Brevibacterium lactofermentum* convert L-glutamate (L-Glu) to D-glutamate (D-Glu) and then to γ-amino butyric acid (GABA) (Figure 5, reviewed by Dicks [4]). Chang et al. [139] have shown that *Lactobacillus rhamnosus* JB-1 changes the expression of GABA receptors (GABARs) in the brain, leading to less anxiety and depressive behavior. Low levels of D-Glu in the brain are also associated with Alzheimer’s disease (AD, Figure 5). Several researchers have reported on the positive effect probiotic lactic acid bacteria have on the protection of neurological pathways and ascribe this to the reduction of inflammation, an increase in immune response, and stimulation of neurotransmission through the metabolism of tryptophan to indole derivatives (Figure 5) in the tryptophan–kynurenine signaling pathway [140,141]. In one study, *Bifidobacterium breve* strain A1 administered to mice injected with amyloid-β prevented cognitive impairment and suppressed the expression of pro-inflammatory and immune-reactive genes in the hippocampus [142]. In several studies, lactobacilli and bifidobacteria prevented the progression of AD in mice [143,144,145], increased expression of neuronal proteolytic pathways [143], and reduced neurodegeneration [145]. Akbari et al. [146] have shown an improvement in cognitive functions and metabolic status of AD patients after treatment with a probiotic containing *Lactobacillus acidophilus*, *Lactobacillus casei*, *Bifidobacterium bifidum*, and *Lactobacillus fermentum.* In another study [147], a multispecies probiotic increased the cell numbers of *F. prausnitzii* and improved the circulation of kynurenine in patients with AD. A higher density of immune cells and antigen processing/presentation markers have been reported present in patients with increased levels of *Faecalibacterium* [148].

Butyrate, produced in the colon by *Bifodobacterium*, *Lactobacillus*, *Lachnospiraceae*, *Blautia*, *Coprococcus*, *Roseburia*, and *Faecalibacterium*, provides energy to epithelial cells [149], suppresses inflammatory responses by down-regulating histone deacetylase EC 3.5.1.98 (also referred to as lysine deacetylase) inhibitors (HDACi) [150], and modifies the integrity of the blood–brain barrier (BBB), which affects the CNS and maturation of microglia [151]. Inhibition of HDACi in the frontal cortex and hippocampus of mice, brought about by the administration of sodium butyrate, has alleviated depressive behavior [152], dementia, and brain trauma [153]. Butyrate, on the other hand, also activates G-protein-coupled receptors (GPCRs, Figure 5) that may cause several neurodegenerative disorders [154] and stimulate regulatory T cells to produce inflammatory cytokines [155]. Low levels of butyrate inhibit GPCRs and interrupt immune or hormonal responses [151]. Alteration of the hormone signals reaches the EECs via the hypothalamic–pituitary–adrenal axis (HPA) [154]. Elevated levels of the neuromodulator acetylcholine (Ach) increase the expression of *bdnf*, encoding the brain-derived neurotrophic factor (BDNF, Figure 5) in the frontal cortex and hippocampus, and stimulate brain development [156]. Low levels of BDNF are associated with depression and anxiety [157]. Neurological disorders may thus be prevented by keeping SCFAs and histone deacetylase (HDAC) at optimal levels. One way of achieving this is to maintain a well-balanced gut microbiome. Monitoring butyrate levels may be considered a reporter of inflammation and cognitive abilities. This would, however, be an indirect way of monitoring brain functions related to mental health. More direct approaches need to be considered, i.e., determining the levels of GABARs, HDACi, Ach, and the expression of *bdnf.*

Neuronal conditions such as Alzheimer’s disease (AD), autism spectrum disorder (ASD), multiple sclerosis (MS), Parkinson’s disease (PD), and amyotrophic lateral sclerosis (ALS) are associated with dysfunctional microglia [158]. Fecal transplants from humans with attention deficit hyperactivity disorder (ADHD), AD, and PD to mice have activated the microglia in the brain and aggravated cognitive and physical impairments [159,160,161]. These findings, along with more evidence of a clear link between microbial dysbiosis and neurodevelopmental, neurodegenerative, and psychiatric disorders such as ASD, schizophrenia, AD, major depressive disorder (MDD), and PD [162,163,164,165] have prompted researchers to have a closer look at the GBA.

To the best of our knowledge, no biomarker tests for AD have been approved by the U.S. Food and Drug Administration (FDA). Commercially available tests are AlzheimAlert™ (Nymox Pharmaceutical Corp., Saint Laurent, QC, Canada); Innotest^®^ assays for microtubule-associated protein (T-tau), phosphorylated tau (P-tau), and amyloid β 42 (Aβ42) (Fujirebio Diagnotics, Malvern, PA, USA); AdMark^®^ cerebrospinal fluid (CSF) analysis; DISCERN™ (Neurodiagnostics, Phoenix, AZ, USA) skin sample fibroblast testing; AD-Detect (Quest Diagnostics, Secaucus, NJ, USA); and Lumipulse^®^ G ß-Amyloid Ratio (1–42/1–40) Test (Fujirebio Diagnostics).

## 5. Multiple Sclerosis

Oxidative stress is triggered by the production of reactive oxygen species (ROS), dysfunctional mitochondria, and damaged cells [166,167,168]. Neurons in the CNS are highly susceptible to oxidative stress (Figure 6), which causes chronic inflammation and ultimately demyelination and neurodegeneration [169,170]. Bacterial metabolites (small molecules released by gut bacteria) are known to alter the levels of multiple sclerosis (MS) [170,171] and may be used as biomarkers in MS. Yoon et al. [172] have shown changes in the levels of SCFAs such as acetate, propionate, and butyric acid in individuals with MS. Studies with mice have shown that SCFAs released from gut microbiota have anti-inflammatory properties through anti-inflammatory cytokine production and restore BBB integrity [99,173,174]. This is summarized in Figure 6. It is, however, important to note that SCFAs have an opposite effect than long-chain fatty acids (LCFAs). Dietary LCFAs promote a pro-inflammatory status through Th17 cell activation. One of the causes of MS is the infiltration of the CNS by peripheral autoreactive immune cells that cross a damaged BBB [175,176]. These autoreactive immune cells (e.g., Th1, Th17, and CD8^+^ T), activate B cells to release autoantibodies [177] (Figure 6). Overactivation of exogenous and endogenous CNS immune cells leads to neuroinflammation and neurodegeneration [178]. This is usually a cascade of events. When autoreactive CD4^+^ T cells invade the CNS, Th17 cells are activated and express chemokine receptor 6 (CCR6) that binds to the CCL20 ligand on endothelial cells in the BBB [179] (Figure 6). The release of IL-17 and granulocyte-macrophage-colony-stimulating factor (GM-CSF) by activated CD4^+^ Th17 cells triggers the activation of microglia and macrophages (Figure 6). Mononuclear phagocytes activated by GM-CSF migrate into the CNS and induce the production of ROS species that damage neurons [180] (Figure 6). CD4^+^ Th1 cells also cross the BBB and release gamma-interferon (IFN-γ) which activates microglia to release IL-12 and initiate a CD8^+^ cytotoxic T cell response [181]. IFN-γ induces the formation of major histocompatibility complex class I molecules (MHC I), which regulates the immune response to pathogens [182]. With the increase in Th1 and Th17 cell levels, pro-inflammatory molecules such as cytokines, interferons, and tumor necrosis factor (TNF) are released, causing further damage to the CNS [183]. IL-17 produced by Th17 and CD8^+^ cells recruits neutrophils and monocytes to the CNS [183].

Th1 and Th17-related pro-inflammatory cytokines, interferons, and tumor necrosis factor (TNF) may thus be possible reporters of MS. The pro-inflammatory environment created by Th1 and Th17 cells attracts peripheral monocytes and promotes neuronal damage. In a normal immune response, T (and B) cells fight microbial infections. B cells produce immunoglobulin A (IgA). Changes in these defense mechanisms lead to gut dysbiosis and the risk of developing autoimmune diseases [184]. In MS, B cells are transformed into CXCR3^+^ cells that infiltrate the brain [185], and central as well as peripheral tolerance mechanisms control the development of autoreactive B cells [186]. MS individuals have a defective peripheral B cell tolerance [187], and activated microglia release pro-inflammatory mediators (e.g., ROS, NO, and peroxynitrite) that are involved in the phagocytosis of myelin, formation of antigen T cells, and production of cytokines [188]. Pro-inflammatory mediators such as interleukins, NO, and ROS cause neurodegeneration. NO produced by activated microglia alters energy production at a cellular level, which leads to increased intracellular Ca^2+^ and damage to neuronal cells. In addition, TNFα promotes the apoptosis or pyroptosis of neuronal cells [189]. Acute demyelinated lesions in MS progress into chronic active inflammatory lesions [190]. Astrocytes maintain BBB function and neurotransmitter levels. They regulate enzymes involved in glutamate production. In individuals with MS the regulatory role of astrocytes is repressed, leading to an increase in glutamate levels, glutamate-mediated excitotoxicity, and neuronal death [189].

Gut microbiota maintains gut barrier integrity [190] but also modulates mitochondrial activity and ROS production, either directly or indirectly through the production of SCFAs and formyl-peptides or the activation of NADPH oxidase [191]. It is thus no surprise that individuals diagnosed with MS suffer from gut dysbiosis. An increase in ROS alters intracellular signaling pathways and promotes inflammation in MS patients [192,193]. ROS also modulates redox signaling pathways in intestinal cells, such as the nuclear factor erythroid 2-related factor (Nrf2) pathway, TLR pathway, or NF-κB pathway. In healthy individuals with a balanced gut microbiome, the interaction between gut bacteria and the immune system protects cells from ROS damage. Nrf2 decreases ROS levels by regulating mitochondrial functions [192]. *Clostridium* spp. and *Lactobacillus rhamnossus* inhibit NFκB-mediated inflammation by producing metabolites that control immune response [193,194,195,196,197]. NF-κB can thus play a protective role in the gut [198]. If NF-κB is at below-optimal physiological levels, as in individuals with MS, mucosal innate immunity is compromised and pathogens increase [197,198]. Tryptamine, produced from tryptophan, decreases inflammation by stimulating the growth of butyrate-producing gut bacteria [199]. A derivative of tryptamine, 5-hydroxy-tryptamine (5-HT or serotonin), acts as an anti-depressant. Low levels of 5-HT are found in MS patients. Serotoninergic modulators such as SSRIs activate microglia and decrease oxidative stress and neuroinflammation [200,201]. N-acetyl-5-methoxytrypatmine (melatonin), another tryptamine derivative, acts as a signaling molecule at the CNS level and regulates immune response, oxidative stress, and apoptosis. However, in a recent review it was mentioned that melatonin may instigate oxidative stress and inflammation [202].

Developing a reporter for MS is a challenge, as the molecule may be produced endogenously or by intestinal microbiota. This is intensified by the high similarity between compounds produced by gut microbiota and human cells. A recently published gut microbiota-specific exposome database [203] may provide answers to the selection of reporters for MS. This would require pooling information associated with MS from databases on exposomes, genes, and metabolites (including those of bacterial origin).

## 6. Diabesity

Diabesity, described as a strong pathophysiological link between diabetes and excess body weight, is becoming a global concern [204] and is treated using pharmacological and non-pharmacological approaches. Fundamental studies have shown that changing the gut microbiome may be one way of treating diabesity. Proposed ways of facilitating such a change include the use of nano nutraceuticals (including nanoprobiotics), nanoprebiotics, and plant-derived nanovesicles (Figure 7). Several studies have shown that an increase in *Bifidobacterium*, *Enterococcus*, *Bacteroides*, *Lactiplantibacillus plantarum* (previously *Lactobacillus plantarum*), and *Akkermansia muciniphila* regulates diabesity [204,205]. Obese individuals have an abundance of Firmicutes, *Alistipes*, *Anaerococcus*, *Fusobacterium*, and *Parvimonas* (Figure 7), whereas lean people have more *Bacteroides*, *Faecalibacterium*, and *Lachnoanaerobaculum* [206]. In general, a healthy gut is dominated by Bacteroidetes and Firmicutes [207]. This has led nutritionists to believe the Firmicutes/Bacteroidetes (F/B) ratio may be an indicator of GIT health. This is confirmed by an increase in F/B amongst obese individuals and the opposite (a decrease) during weight loss [208]. Lean people have a high population of bacteria associated with anti-inflammatory properties, whereas obese people have more bacteria with pro-inflammatory properties [206] (Figure 7). Similar changes in bacterial populations are observed in people with diabetes. The onset and progression of diabetes are characterized by an increase in pro-inflammatory bacteria and a decrease in anti-inflammatory bacteria [209]. *Eubacteria rectale*, *Roseburia*, *Verrucomicrobia*, *Clostridium*, *F. prausnitzii*, and *Akkermansia muciniphila* inhibit the effects of diabetes, whereas *E*. *coli*, *Bacteroides stercoris*, *Desulfovibrio*, *Clostridium mutans*, *Streptococcus mutans*, *Lactobacillus gasseri*, and *Haemophilus parainfluenza* promote the effects of diabetes [210,211] (Figure 7). An imbalance in gut microbiota triggers inflammatory responses and affects insulin-related signaling pathways, such as the mammalian target of the rapamycin signaling pathway and TLR4/NF-κB signaling. This leads to insulin resistance and elevated blood glucose [212,213]. Currently, there is no specific species that can be used as a reporter for diabesity. Even if such a species is identified, further research is required to determine the threshold of cell numbers considered to be a warning sign for developing diabesity. As with many other abnormalities, more reliable reporters would be enzymatic profiling or the overexpression of specific genes by gut microbiota.

## 7. Stroke

Ischemic stroke (IS) is described as a blockage of blood supply to part of the brain, causing irreversible damage and brain tissue necrosis [214]. Half of the individuals who have had IS suffer from gastrointestinal complications, including dysphagia, gastrointestinal bleeding, constipation, and intestinal incontinence [215] (Figure 8). It is interesting to note that people with Alzheimer’s disease, Parkinson’s disease, stroke, and autonomic spectrum disorder have an imbalanced gut microbiome [216,217,218,219]. The GIT Firmicute and Bacteroidetes cell numbers decrease in individuals who have had a stroke, while the Proteobacteria increase (Figure 8). The difference between these phyla correlates with the severity of the stroke. The increase in trimethylamine (TMA)-producing bacteria of people who have had a stroke (Figure 8) is ascribed to the abundance of dietary quaternary ammonium compounds, mainly choline [220,221]. Such a change in bacterial phyla may serve as a stroke warning, thus a reporting system may be referred to, e.g., the Stroke Dysbiosis Index (SDI) [222]. A high SDI correlates with an increase in Enterobacteriaceae and Parabacteroides but a decrease in *Fecalibacterium*, *Clostridiaceae*, and *Lachnospira* (Figure 8). The value of using an SDI as a reporter system for stroke has been demonstrated in a murine experiment. Mice that received fecal transplants from patients with a high SDI experienced severe brain damage, increased levels of IL-17 and T cells (Figure 8), and a significantly higher risk of stroke than mice that received normal fecal transplants [203]. Changes in the gut microbiome from beneficial to pathogenic phyla lead to an increase in endotoxins such as LPS that ultimately enter the circulatory system. Endotoxins stimulate the overexpression of Toll-like receptors (TLRs), and activate the nuclear factor kappa B (NF-κB) pathway, peripheral immune response, and chronic inflammatory response [223] (Figure 8). An imbalanced gut microbiome can also increase the levels of oxidized low-density lipoprotein (OX-LDL), causing blood vessel constriction and, thus, hypertension [224]. Blood pressure is regulated by SCFAs produced by gut microbiota, via the gut-sympathetic nervous system axis [225]. Apart from the GBA and HPA, the nervous and immunological pathways, the gut–blood (intestinal epithelial and mucosal) barrier, and the BBB play a major role in brain functions [226].

## 8. Conclusions

Bacterial interactions with the human GIT are difficult to study, not only due to the vast number of coexisting species found in the microbiome but also due to the difficulty in culturing many of the bacteria. Interactions between intestinal bacteria and human cells may occur randomly without selective pressure or develop due to changes in immune responses caused by certain diseases. Further research is required to understand the interactions between bacterial proteins and proteins of human origin and the effect small molecules such as SCFAs have on gut homeostasis, regulation of the immune system, maintenance of IECs, and regulation of neurological and endocrine functions. Other physiological functions may also be modulated and regulated.

Abnormalities such as ARLD, cancer, cognitive impairment, MS, diabesity, and IS may not always be preventable, but the early detection of these diseases is important and affects the success of treatments. Although diagnostic tests are available to identify these diseases, most are based on changes in certain enzymatic and chemical profiles. Some of these early screening tests are unreliable, as seen with the monitoring of serum aminotransferase levels in the diagnosis of ARLD. In this case, a combination of biomarkers such as CDT, GGT, and MCV have to be used to validate the results. Alternative options need to be investigated, such as using REG3B and REG3G as reporter molecules to detect early changes in gut epithelial Muc2, which is strongly associated with ARLD. Changes in the levels of the circadian regulator Per2 or expression of *Per2*, and levels of the fibroblast growth factor FGF-19 may also serve as reporters in the early detection of ARLD. The early detection of colon cancer is possible by testing for colibactin and SMO and changes in butyrate levels. The importance of monitoring butyrate levels is also important from an immunological perspective. With a decline in butyrate, HDACi levels increase, leading to an increase in depressive behavior, dementia, and other brain traumas. At the same time, GPCRs are inhibited, causing immune and hormonal imbalances. Butyrate may thus be a reporter of inflammation, cognitive abilities, and mental health. Pro-inflammatory molecules such as cytokines, interferons, and TNF may be used as early reporters of MS. These are just a few examples of microbial enzymes and metabolites that may be considered in the development of early reporter systems.

## Figures and Tables

**Figure 1 ijms-25-04431-f001:**
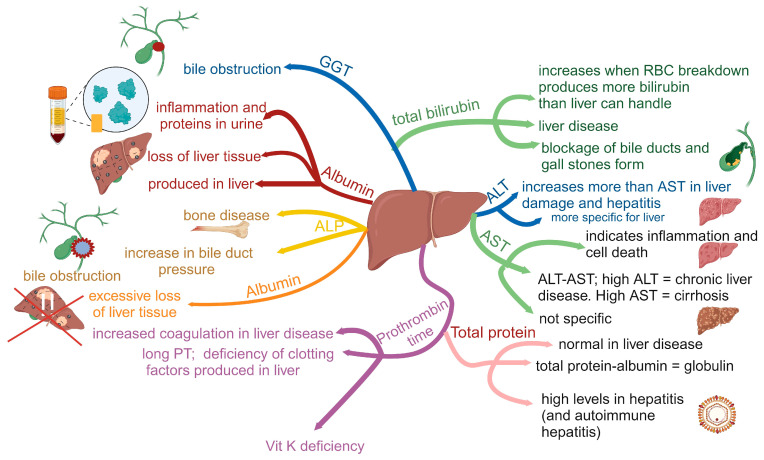
Biomarkers that are normally used in liver function tests. ALP = alkaline phosphatase, PT = prothrombin time, GGT = gamma-glutamyl transferase, AST = aspartate aminotransferase, ALT = alanine aminotransferase. An AST:ALT ratio above 2 is normally considered a reliable indication of ARLD. The red cross denotes a loss in liver tissue. Constructed using Biorender.com (25 March 2024).

**Figure 2 ijms-25-04431-f002:**
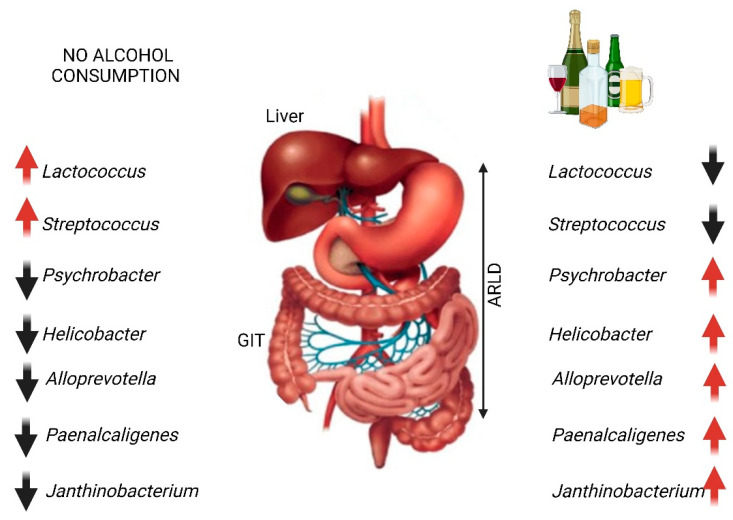
Changes in some bacterial populations when alcohol is consumed. GIT = gastrointestinal tract, ARLD = alcohol-related liver disease. Changes in cell numbers of some genera may be symptomatic of ARLD. Red arrows denote an increase in cell numbers and black arrows a decrease. Constructed using Biorender.com (25 March 2024).

**Figure 3 ijms-25-04431-f003:**
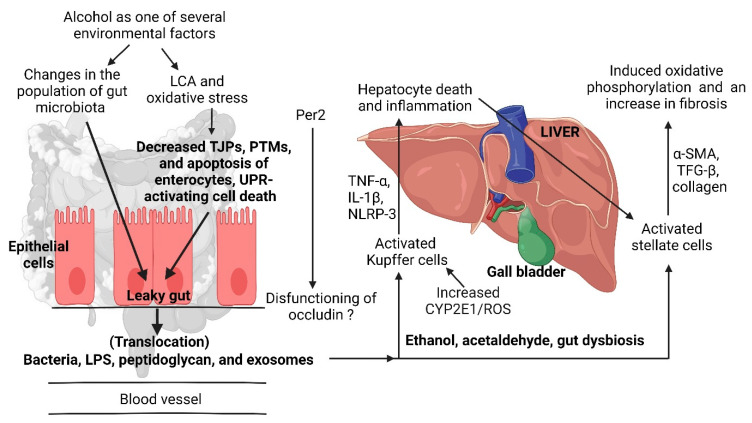
Damage caused by high alcohol consumption. An elevation in CYP2E1 and NADPH oxidases (NOXs) increases oxidative stress. Gut dysbiosis alters gut metabolism and damages the intestinal barrier (epithelial cells) through various mechanisms, including oxidative stress-mediated post-translational modifications (PTMs) that decrease the interactions between tight junction proteins (TJPs). The release of fibroblast growth factor 19 (FGF-19) into the portal vein inhibits the synthesis of hepatic bile acid (BA) in the liver. Per2 = period circadian regulator, LCA = lithocholic acid, LPS = lipopolysaccharide, TNF-α = tumor necrosis factor alpha, IL-1β = isoleucine 1 beta (a pro-inflammatory cytokine), NLRP-3 = Nod-like receptor protein 3, α-SMA = alpha smooth muscle actin, TFG-β = transforming growth factor beta, CYP2E1 = cytochrome P450 2E, ROS = reactive oxygen species. Created using Biorender.com (25 March 2024).

**Figure 4 ijms-25-04431-f004:**
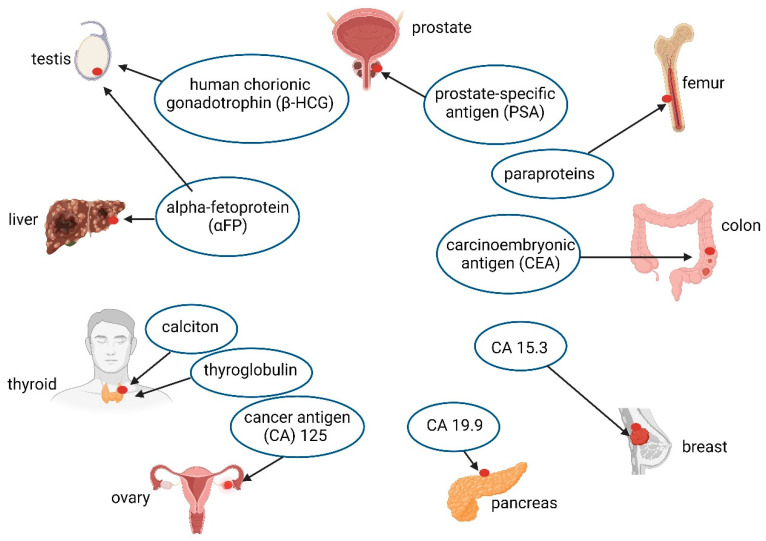
Some biomarkers currently used to detect prevalent cancers. Created using Biorender.com (11 April 2024).

**Figure 5 ijms-25-04431-f005:**
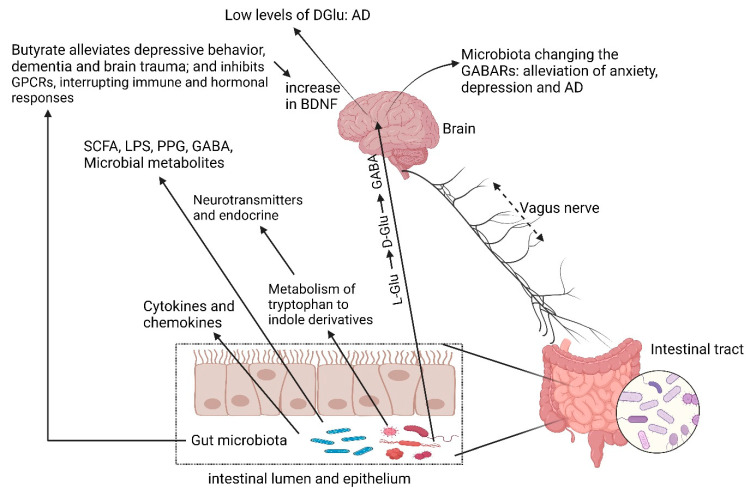
Cognitive functions are influenced by metabolites produced by gut microbiota. A few of these metabolic compounds are listed here. SCFA = short-chain fatty acids, LPS = lipopolysaccharides, PPG = peptidoglycans, GABA = γ-amino butyric acid, GPCR = G protein-coupled receptor, AD = Alzheimer’s disease, GABARs = GABA receptors, BDNF = brain-derived neurotrophic factor, L-Glu = L-glutamine, D-Glu = D-glutamine. Dysbiosis can be caused by stress that may alter tryptophan levels, SCFA levels, the immune system, and gut permeability. The release of cytokines and chemokines (IL-6, IL-1β, IL-8) can lead to neuroinflammation and activation of the hypothalamic–pituitary–adrenal (HPA) axis. Created using Biorender.com (11 April 2024).

**Figure 6 ijms-25-04431-f006:**
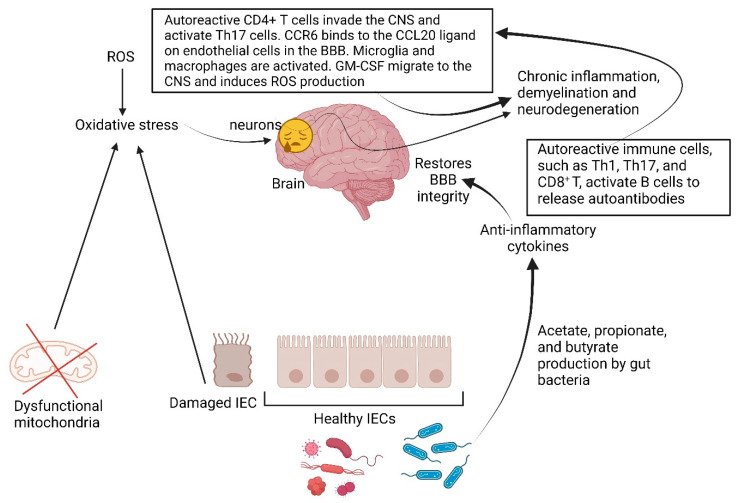
A summary of reactions that cause multiple sclerosis (MS). CNS = central nervous system, CCR6 = chemokine receptor 6, BBB = blood–brain barrier, GM-CSF = granulocyte-macrophage-colony-stimulating factor, ROS = reactive oxygen species, IEC = intestinal epithelial cell. The red cross denotes dysfunctional mitochondria. Created using Biorender.com (12 April 2024).

**Figure 7 ijms-25-04431-f007:**
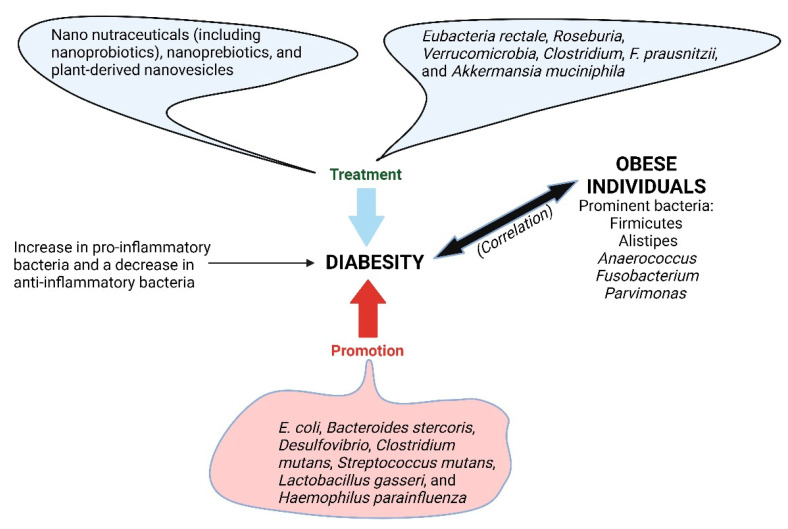
Diabesity, described as a strong pathophysiological link between diabetes and excess body weight, is regulated by gut microbiota. Created using Biorender.com (12 April 2024).

**Figure 8 ijms-25-04431-f008:**
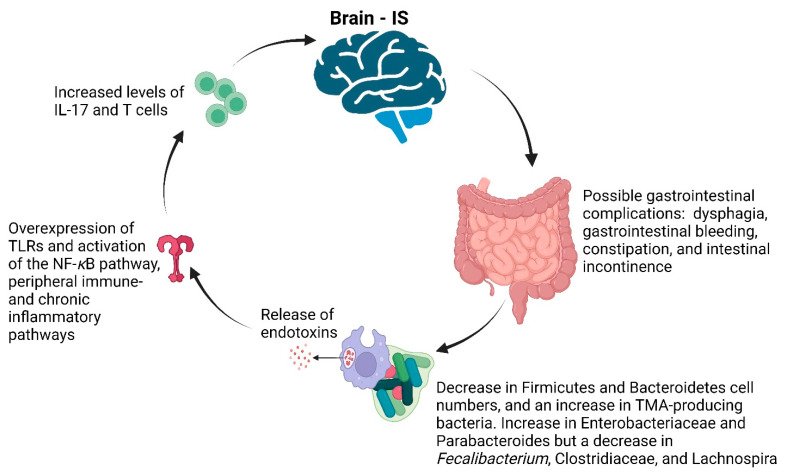
Complications associated with ischemic stroke (IS). TMA = trimethylamine, TLRs = Toll-like receptors, NF-κB = nuclear factor kappa B, IL-17 = isoleucine 17. Created using Biorender.com (12 April 2024).

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
