# Peer review of "Gut Bacteria Provide Genetic and Molecular Reporter Systems to Identify Specific Diseases"

_ijms, 2024, doi:10.3390/ijms25084431_

Round 1

Reviewer 1 Report

Comments and Suggestions for Authors

This manuscript contributes to the field and is interesting!

1. Correct all the figures, it seems that the writing and images overlap, so that the writing is not visible and cannot be read.

2. Points 3, 4, 5, 6, and 7 really need figures that can make it easier for readers to understand, or at least a table.

Author Response

This manuscript contributes to the field and is interesting!

  1. Correct all the figures, it seems that the writing and images overlap, so that the writing is not visible and cannot be read.

Answer: The Figures have all been corrected so that the writing does not overlap with the images (see Figs 1 and 3).

  1. Points 3, 4, 5, 6, and 7 really need figures that can make it easier for readers to understand, or at least a table.

Answer:  Figures have now been added for the five sections mentioned (numbered as Figures 4 to 8). All figures were set at 600dpi.

Reviewer 2 Report

Comments and Suggestions for Authors

Please review the article titled "Gut Bacteria Provide Genetic and Molecular Reporters" for publication in IJMS. Overall, it is a promising paper with a few areas that could be enhanced:

  1. Abstract Introduction: Consider reworking the introduction to the abstract to provide a clearer overview of the paper's focus without delay.

  2. Introduction Expansion: The final paragraph of the introduction should be expanded to clearly articulate the main objectives of the article.

  3. Introduction Length: The introduction could be slightly expanded given the breadth of the topic.

  4. Alcohol Discussion: While acknowledging the harmful aspects of alcohol, it might be beneficial to discuss its positive properties and appropriate dosage considerations.

  5. Figure 1 Quality: The quality of Figure 1 could be improved; please consider enhancing its clarity.

  6. Figure 3 Quality: Similarly, the quality of Figure 3 needs improvement; ensure it is of higher resolution.

  7. Refinement of Sentences: The sentence regarding the use of colibactin and SMO as reporters of colon cancer is quite general; refine this statement to provide more specificity.

  8. Cancer Causation Clarification: Regarding cancers not caused by direct contact with microbial cells or TME (tumor microenvironment), please clarify this point further.

Overall, the manuscript is strong, and with these adjustments, I fully support its publication.

Author Response

Abstract Introduction: Consider reworking the introduction to the abstract to provide a clearer overview of the paper's focus without delay.

Answer: The abstract has been rewritten (lines 10-29) to provide a clearer overview of the paper’s focus.

Introduction Expansion: The final paragraph of the introduction should be expanded to clearly articulate the main objectives of the article.

Introduction Length: The introduction could be slightly expanded given the breadth of the topic.

Answer: A section has been worked into the final paragraph of the introduction (lines 111-116).  Additional information has also been worked into the introduction (lines 56-92), as also requested by one of the other reviewers.

Alcohol Discussion: While acknowledging the harmful aspects of alcohol, it might be beneficial to discuss its positive properties and appropriate dosage considerations.

Answer: This has now been addressed (see lines 125-128).

Figure 1 Quality: The quality of Figure 1 could be improved; please consider enhancing its clarity.

Figure 3 Quality: Similarly, the quality of Figure 3 needs improvement; ensure it is of higher resolution.

Answer: The quality of both these figures has been improved. All Figures are set at 600 dpi.

Refinement of Sentences: The sentence regarding the use of colibactin and SMO as reporters of colon cancer is quite general; refine this statement to provide more specificity.

Answer: The text has been amended to address this point (lines 301-305).

Cancer Causation Clarification: Regarding cancers not caused by direct contact with microbial cells or TME (tumor microenvironment), please clarify this point further.

Answer: This has been clarified (lines 319-322).

Overall, the manuscript is strong, and with these adjustments, I fully support its publication.

Reviewer 3 Report

Comments and Suggestions for Authors

Thank you for inviting me to review this paper on the role of gut bacteria in extraintestinal diseases. I found the review to be well-written, providing a comprehensive and detailed analysis of the topic. Overall, I do not identify major shortcomings and recommend accepting it with minor revisions.

One suggestion I have is to include references to the role of the microbiota in gastrointestinal diseases, such as irritable bowel syndrome or gastroparesis, in the introduction. This addition would provide context and enhance the comprehensiveness of the review. I recommend briefly discussing or mentioning the role of the microbiota in gastrointestinal diseases, citing relevant papers such as PMID 37630649, 37317096, 35125827, and 31789724.

Thank you once again for the opportunity to review this insightful paper.

Author Response

Thank you for inviting me to review this paper on the role of gut bacteria in extraintestinal diseases. I found the review to be well-written, providing a comprehensive and detailed analysis of the topic. Overall, I do not identify major shortcomings and recommend accepting it with minor revisions.

One suggestion I have is to include references to the role of the microbiota in gastrointestinal diseases, such as irritable bowel syndrome or gastroparesis, in the introduction. This addition would provide context and enhance the comprehensiveness of the review. I recommend briefly discussing or mentioning the role of the microbiota in gastrointestinal diseases, citing relevant papers such as PMID 37630649, 37317096, 35125827, and 31789724.

Answer: Information regarding IBS, gastroparesis and endometriosis has been worked into the introduction (see lines 56-94). An additional 19 papers, including the four mentioned, have been cited and added to the reference list.  All references have been renumbered.

Thank you once again for the opportunity to review this insightful paper.